

# Characterization of *Amaranthus* species: ability in nanoparticles fabrication and the antimicrobial activity against human pathogenic bacteria

Walaa A. Hassan[1], Afrah E. Mohammed[2], Najla A. AlShaye[2], Hana Sonbol[2], Salma A. Alghamdi[2], Duilio Iamonico[3] and Shereen M. Korany[4]

[1] Botany and Microbiology Department, Faculty of Science, Beni-Suef University, Beni-Suef, Egypt
[2] Department of Biology, College of Science, Princess Nourah bint Abdulrahman University, Riyadh, Saudi Arabia
[3] Garden of the Apennine Flora of Capracotta, Piazza Stanislao Falconi, Capracotta, Italy
[4] Helwan University, Botany and Microbiology Department, Faculty of Science, Cairo, Egypt

Corresponding author
Najla A. AlShaye,
naaalshaye@pnu.edu.sa

## ABSTRACT

The present work aimed at differentiating five *Amaranthus* species from Saudi Arabia according to their morphology and the ability in nanoparticle formulation. Biogenic silver nanoparticles (AgNPs) were synthesized from leaf extracts of the five *Amaranthus* species and characterized by different techniques. Fourier-transform infrared spectroscopy (FT-IR) was used to identify the phyto-constituents of *Amaranthus* species. The nanoparticles (NPs) were characterized by UV-visible spectroscopy, dynamic light scattering (DLS), transmission electron microscopy (TEM), and energy-dispersive X-ray spectroscopy (EDX). The antibacterial activity of the synthesized NPs was tested against *Staphylococcus aureus*, *E. coli*, *Klebsiella pneumoniae* and *Pseudomonas aeruginosa* using the agar well diffusion method. Spherical NPs varying in size and functional groups from the five plant species were demonstrated by TEM, DLS and FTIR analysis, respectively. Variations in NPs characteristics could be related to the phytochemical composition of each *Amaranthus* species since they play a significant role in the reduction process. EDX confirmed the presence of Ag in plant fabricated AgNPs. Antibacterial activity varied among the species, possibly related to the NPs characteristics. Varied characteristics for the obtained AgNPs may reflect variations in the phytochemical composition type and concentration among *Amaranthus* species used for their fabrication.

## INTRODUCTION

The genus *Amaranthus* (Caryophyllales Berch. & J. Presl, *Amaranthaceae* Juss.) includes 65–70 species distributed worldwide, half of them are native to America (*Hernández-Ledesma et al., 2015*; *Iamonico, 2015*). Several *Amaranthus* species are used as ornamental plants, as food, or as medicines. They are able to escape from cultivation causing a negative impact on agricultural systems, mainly due to their high seed production and resistance

to herbicides, as reported for some species (*Das, 2016*; *Iamonico, 2010*; *Tranel & Trucco, 2009*). Additionally, they are a low-cost source of protein, minerals, and vitamins; they represent a source of nutrition for many centuries in Asia, Africa, Central, and South America. Furthermore, the genus *Amaranthus* has a diverse genetic pool and has the ability to withstand drought stress, making it an ideal crop for ensuring nutritional security in a climate-changing planet (*Jamalluddin et al., 2021*). From a medical perspective, the crude extract of *Amaranthus* plants contains alkaloids (betacyanins and betaxanthin), polyphenols (flavonoids, steroids, catechuic acid, and tannins), terpenoids (cerasinone and norecasantalic acid), and saponins depending on the plant maturity stage, cultivar type, and geographical location (*Peter & Gandhi, 2017*). Each of these compounds has variable biological activities, including anticancerous, antioxidant, immunomodulatory, and antimicrobial, which inspired researchers all over the world their investigation (*Gandhi et al., 2021*).

*Amaranthus* was originally described in the 1st edition of *Species Plantarum* (*Linnaeus, 1753*), that showed a difficulty in identification of its various species due to high phenotypic variability and hybridization, which resulted in nomenclature and name misapplications (*Assad et al., 2017*; *Costea, Sanders & Waines, 2001*; *Iamonico, 2016a*; *Iamonico, 2016b*; *Iamonico, 2020a*; *Iamonico, 2017*). In fact, many studies revealed that many names are actually heterotypic synonyms (*Iamonico, 2014a*; *Iamonico, 2014b*; *Iamonico, 2020b*; *Iamonico, 2020c*; *Iamonico & Palmer, 2020*).

According to *Chaudhary (1999)*, the flora of Saudi Arabia includes eight *Amaranthus* species. However, based on our ongoing study (*Hassan et al., 2022*), they are actually 12 species (16 taxa including varieties), one of them is *A. tricolor* L., which is only cultivated. But, the presence of another one (*A. graecizans*. subsp. *thellungianus* (Nevski) Gusev) is still doubtful.

Several tools based on morphological, numerical and molecular studies have been used for taxonomic identification of plant species (*Haider, 2018*). However, nanoparticles mediated by plant extracts have tremendous application in many ways of nanotechnology due to its low cost and environmental favorable effect.

Several studies have been undertaken on green synthesis of NPs using plant leaf extracts that are capable of forming silver NPs (AgNPs) with varying sizes and shapes depending on many factors, such as plant type and origin, soil and climate conditions (*Mohammed Afrah et al., 2021*; *Mondal et al., 2011*). Due to the significance of employing plant extract to create silver nanoparticles that differs in their characteristics in relation to the plant type, therefore, varied plant species may have different abilities in nanoparticles (NPs) fabrication suggesting varied phytochemicals and varied NPs properties accordingly.

This special capability of green fabrication of NPs inspired our research group to use this modern technique to differentiate among angiosperms, based on the plant's ability to synthesize NPs of various sizes, shapes, and concentrations. Previous taxonomic study using FTIR technique was based on AgNPs synthesized from the extracts of eight *Launaea* species (*Zareh et al., 2018*). The current work represents a pioneer attempt to study AgNPs properties in relation to the *Amaranthus* sp. applied for their fabrication with reference to their bio-reducing and capping abilities which may offers AgNPs with varied

characteristics. Such variation could be expected since varied species may have varied phytoconstituents that highly affect the NPs characteristics (*Alharbi, Abaker & Makawi, 2023*). The selected species existing in Saudi Arabia were as follows: *A. blitoides* S. Watson s.str. (naturalized alien in Saudi Arabia), *A. blitum* L. s.str. (naturalized), *A. dubius* Mart. ex Thell. (casual), *A. graecizans*. s.str. (native), and *A. viridis* L. (naturalized). The fabricated NPs were characterized using different techniques and their antimicrobial activity against some pathogenic bacteria was tested.

# MATERIALS AND METHODS

## Morphology

Five taxa of *Amaranthus* (*A. blitoides* var. *blitoides*, *A. blitum* subsp. *blitum* var. *blitum*, *A. dubius*, *A. graecizans* subsp. *graecizans*, and *A. viridis*) were collected from Saudi Arabia (Fig. 1) during the spring of 2021. They were identified and deposited in the herbaria PNUH and RO (*Thiers, 2022*). High resolution images of the synflorescences were obtained using the Leica IC80 HD photo camera and Leica Application Suite program, version 4.5.0. (Leica, Wetzlar, Germany). The images were later processed using Helicon Focus, version 6.6.1 Pro where different focus of the same sample were merged together.

The map was prepared using ArcGIS programme (Esri, Redlands, California, USA).

## Nanoparticles investigation

### Preparation of Amaranthus spp leaf extract

Leaves from the five *Amaranthus* species were collected and washed with double distilled water to remove the adhering substance before being cut into small pieces. The leaf extracts were prepared by adding 100 ml of distilled water to the washed leaves and heating at 80 °C for 10 min. The leaf extracts were filtered through Whatmann No. 1 filter paper. The leaf extract of each species was used separately for AgNPs synthesis and the filtrate was stored in the refrigerator at 4 °C for further use (*Muthukumar et al., 2020*).

### Biosynthesis of silver nanoparticles

Aqueous leaf extracts were added separately to 1 mM AgNO$_3$ at a ratio of 1:10 (v:v) in dark pottles and heated for 15 min at 80 °C. The reaction was kept in the dark at room temperature for 24 h until the developed color was stable. The mixture was centrifuged for 15 min at 14,000 rpm, the supernatant was removed, and the pellet was washed twice with distilled water. The pellet was allowed to dry at room temperature for further analysis, as described before (*Mohammed Afrah et al., 2021*)

### Characterization of Amaranthus species synthesized

Characterization of the biogenic synthesized silver nanoparticles (AgNPs) using *Amaranthus* species was carried out with the following techniques:

*Ultraviolet–visible spectroscopy.* After the dark brown color became stable, ultraviolet–visible (UV–Vis) spectroscopy absorption was determined using a spectrophotometer (UVProbe software, UV-1800; Shimadzu Corporation, Kyoto, Japan). The absorption

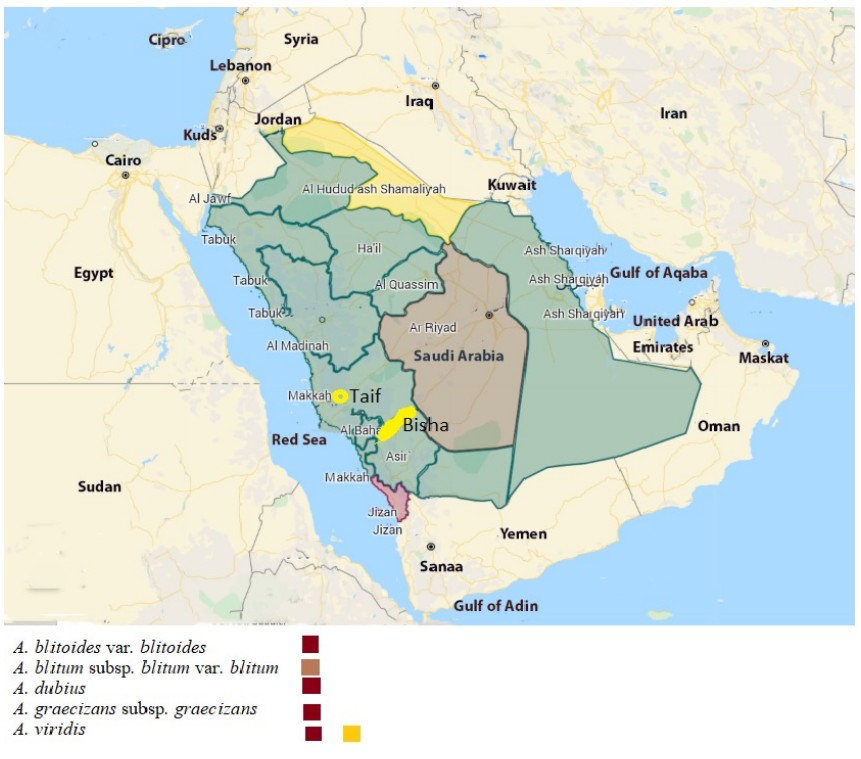

A. blitoides var. blitoides
A. blitum subsp. blitum var. blitum
A. dubius
A. graecizans subsp. graecizans
A. viridis

**Figure 1** **Map of Saudi Arabia showing the distribution of *Amaranthus* species used in this study.** The map was prepared using ArcGIS programme (map credit: Esri, Redlands, California, USA).

spectra of the biosynthesized NPs solution were measured in the range of 200–600 nm after 24 h of reaction, where plant extracts were used as a blank (*Dasari et al., 2013*).

*Fourier transform infrared (FTIR) spectroscopy.* The FTIR spectrum was performed to detect the potential of biomolecules in the *Amaranthus* spp. leaf extracts to reduce AgNO$_3$. The SPECTRUM100 (Perkin-Elmer, Shelton, CT, USA) was used for this purpose and the data scanning was done at a range of 450–3,500 cm$^{-1}$ using a diffuse reflectance accessory, as described previously (*Siddiqi et al., 2018*).

*Hydrodynamic size and surface potential analysis.* The dynamic light scattering (DLS) method was used to determine the pattern of the size distribution. A Zetasizer (NANO ZSP, ver 7.11, Malvern Instruments Ltd, Malvern, UK) was used for this measurement, as described by *Paul, Singh & Sasikumar (2015)*.

*Morphology and size distribution analysis using transmission electron microscopy (TEM).* Determination of the size distribution and morphology of the synthesized AgNPs was performed by the transmission electron microscope (TEM) model JEM-1011 (JEOL, Tokyo, Japan) at 80 kV. A drop of AgNPs solution was applied on the carbon-coated copper grid (200 mesh). The size of particles was calculated using ImageJ 1.45 s software1493, as described previously (*Mie et al., 2014*).

*Energy-dispersive X-ray spectroscopy (EDS).* The JED-2200 series (JEOL) scanning electron microscope (SEM) supplied with energy-dispersive X-ray spectroscopy (EDS) was used to confirm the presence of the silver element by surface analysis of the biosynthesized NPs, as described elsewhere (*Paul, Singh & Sasikumar, 2015*)

## Antimicrobial activity of the *Amaranthus* spp.-based silver nanoparticles

The antibacterial activity of biosynthesized AgNPs was performed *in vitro* against one Gram-positive bacteria, *Staphylococcus aureus* (MW846278) and three Gram-negative bacteria, which were *Pseudomonas aeruginosa* (MW846270), *Escherichia coli* (MW846276) and *Klebsiella pneumoniae* (MW846277) using the agar well diffusion assay, as described by *Awwad, Salem & Abdeen (2013)*. The bacterial strains were obtained from the Bio-house Medical Lab (Riyadh, Saudi Arabia). Briefly, Oxoid nutrient agar medium plates were streaked with the suspension of test bacterial strain ($2.5 \times 10^5$ cfu/ml) and left to dry before making the wells. To each well on the seeded agar plate, 20 ul of AgNPs, plant aqueous extracts and AgNO$_3$ were added separately, and ampicillin was included as a positive control. The plates were incubated at 37 °C for 24 h and the inhibition zones around wells were measured in millimetres. The assay was performed in triplicate, according to the guidelines of the Clinical and Laboratory Standards Institute (*Wayne, 2010*).

## Statistical analysis

Statistical analysis and the antimicrobial data (means $\pm$ standard deviations) beside the figure were produced using GraphPad PRISM 9.1 (San Diego, CA, USA). One-way and two-way ANOVA (Tukey's multiple comparisons and Fisher's LSD) were applied to analyse the differences among the antimicrobial activities of all AgNPs against each bacterial strain.

# RESULTS

## Morphology

The five *Amaranthus* taxa differed from each other in synflorescence characters (general structure), perianth (number of tepals), and fruit (surface). *A. dubius* differed from the other four taxa by the number of tepals (five). *A. blitoides* was characterized by a perianth with four tepals and leaves with a white marginal vein. *A. graecizans* s.str., *A. blitum* s.str., and *A. viridis* had three tepals. *A. graecizans* displayed synflorescences in axillary glomerules, whereas the other two taxa had synflorescences arranged in spike- or panicle-like structures. The difference between *A. viridis* and *A. blitum* s.str. was in the fruit, which was smooth, or slightly rugose on the surface and strongly rugose, respectively (Fig. 2).

## Characterization of the nanoparticles prepared by *Amaranthus* species
### *Ultraviolet–visible spectroscopy*

The fabrication of Ag ions into AgNPs was achieved by the five species of *Amaranthus* providing B-AgNPs, G-AgNPs, D-AgNPs, L-AgNPs, and V-AgNPs for those prepared by *A. blitum* subsp. *blitum* var. *blitum*, *A. graecizans*. subsp. *graecizans*, *A. dubius*, *A. blitoides* var. *blitoides*, and *A. viridis,* respectively. AgNPs formation was first detected by gradual change in the mixture color from colorless to stable brown after 24 h for all tested plant

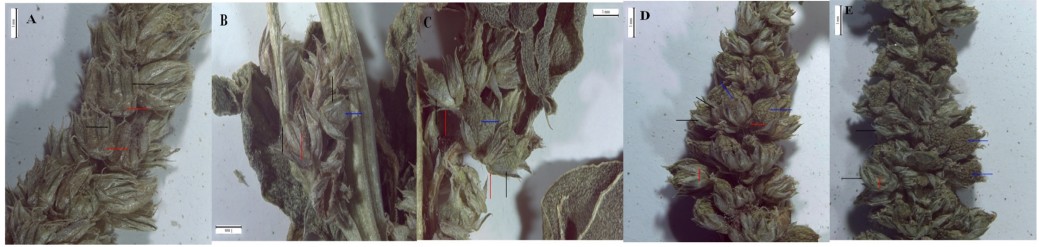

**Figure 2** The images of the synflorescences of; (A) *A. dubius*, (B) *A. blitoides* (C) *A. graecizans*, (D) *A. blitum*, (E) *A. viridis.* Where red arrows refer to bracts, black arrows refer to tepals and blue arrows refer to the fruit.

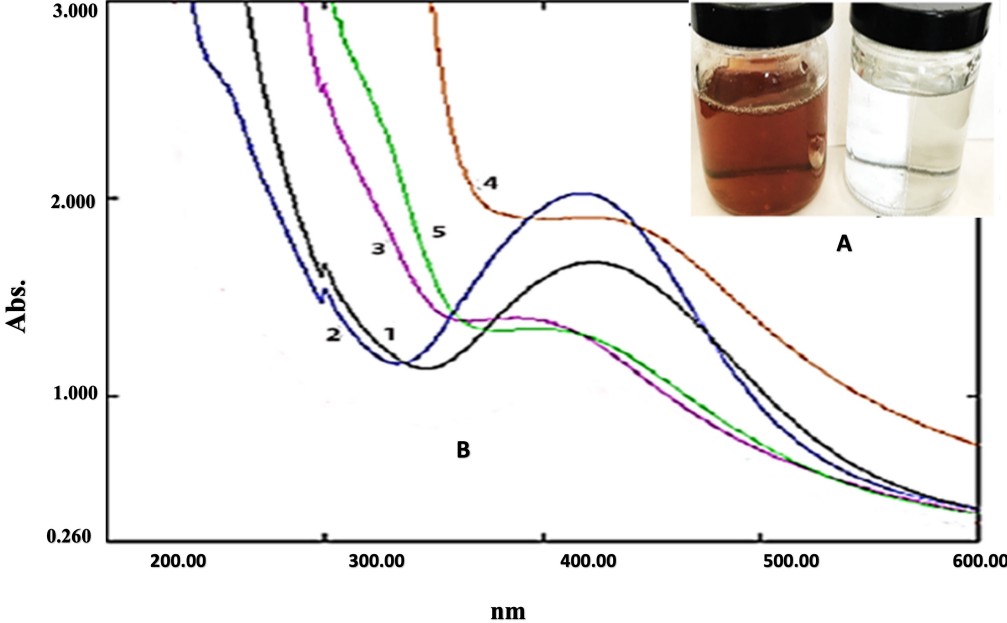

**Figure 3** Ultraviolet–visible spectroscopy results of biosynthesis of AgNPs using leaf extracts of five *Amaranthus* species. (A) shows the change in color of AgNO$_3$ from colorless to brown after addition of leaf extracts. (B) the UV/vis spectrum for: (1) B-AgNPs, (2) G-AgNPs, (3) D-AgNPs, (4) L-AgNPs (4), and (5) V-AgNPs.

species (Fig. 3A), which was further confirmed by UV spectra that indicated wavelengths ranging from 400 to 450 nm (Fig. 3B). A slight variation in the time needed for color change among the tested species was noted.

### Fourier transform infrared spectroscopy

The Fourier transform infrared (FTIR) spectroscopy was applied to demonstrate the functional groups involved in AgNPs fabrication from the five *Amaranthus* species in comparison with the aqueous plant extracts utilized for their synthesis (Fig. 4). Variations in the FTIR peak values were noted between those of the plant extracts and those of

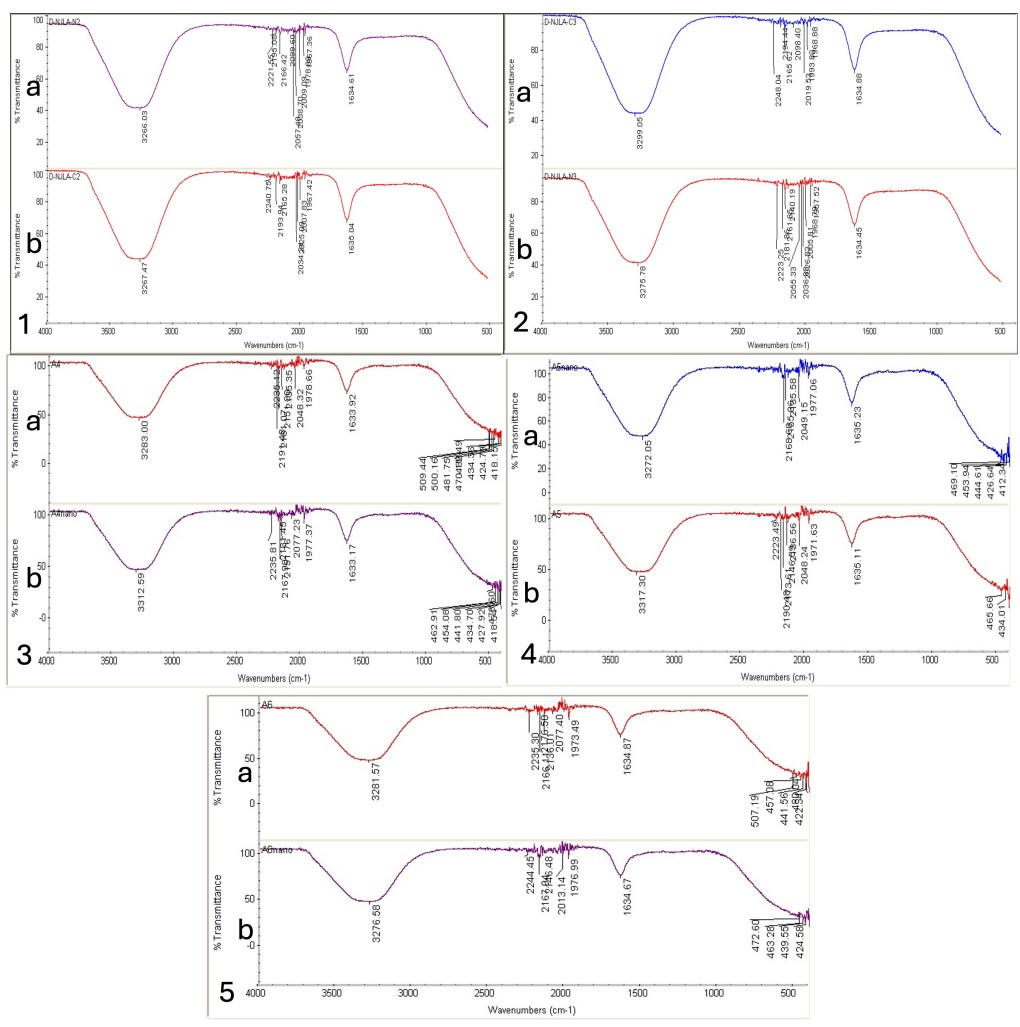

**Figure 4** **FTIR absorbance peaks of aqueous extracts the *Amaranthus* species (A) and the corresponding fabricated AgNPs (B).** *A. blitum* subsp. *blitum* var. *blitum* (1), *A. graecizans* L. subsp. *graecizans* (2), *A. dubius* (3), *A. blitoides* var. *blitoides* (4), and *A. viridis* (5).

plant extract-based NPs for the five *Amaranthus* species, indicating slight differences in the functional groups. Peaks of strong absorbance for *A. blitum* subsp. *blitum* var. *blitum* extract was recorded at 1,635.04 and 3,267.47 cm$^{-1}$, while those for B-AgNPs were 1,634.61 and 3,266.03 cm$^{-1}$. *A. graecizans*. subsp. *graecizans* extract recorded two strong peaks at 1,634.88 and 3,299.05 cm$^{-1}$, while 1,634.45 and 3,275.78 cm$^{-1}$ were recorded for G-AgNPs. The signals of *A. dubius* extract were at 1,633.92 and 3,283 cm$^{-1}$, but moved to 1,633.27 and 3,312.59 cm$^{-1}$ for D-AgNPs. *A. blitoides* var. *blitoides* extract signals were at 1,635.11 and 3,317.30 cm$^{-1}$ that changed with L-AgNPs to 1,634.23 and 3,272.05 cm$^{-1}$. The signals at 1,634.87 and 3,281.57 cm$^{-1}$ were noted for *A. viridis* extract, which changed to 1,634.67 and 3,276.58 cm$^{-1}$ in V-AgNPs.

### Hydrodynamic size and surface potential analysis

The DLS system results showed variation in AgNPs sizes of the NPs produced by the five *Amarathus* species. Sizes of 64.24, 185.1, 72.93, 123.1, and 77.52 nm and polydispersity indexes (PDI) of 0.19, 0.23, 0.20, 0.14, and 0.14 were recorded for B-AgNPs, G-AgNPs, D-AgNps, L-AgNPs and V-AgNPs, respectively (Fig. 5). Following the same sequences of the AgNPs type, the zeta potential values were $-33$, $-23.7$, $-18.3$, $-28.5$, and $-14$, respectively (Fig. 6). Generally, the AgNPs prepared by the five *Amaranthus* species showed variations in NPs characteristics, as shown in Table 1.

### Morphology and size distribution analysis using transmission electron microscopy and energy-dispersive X-ray spectroscopy

The transmission electron microscopy (TEM) imaging for synthesized AgNPs from the five *Amaranthus* species showed mostly spherical NPs. Areas around AgNPs represent clear capping agents from the plant extract (Fig. 7).

The EDX analysis confirmed the presence of the silver element in all synthesized NPs, besides the carbon and oxygen that originated from the plant extract (Fig. 7). The results demonstrated silver signals at 3 keV, along with carbon and oxygen peaks for all synthesized AgNPs.

## Determination of the antibacterial activity of AgNPs

Inhibition of the bacterial growth by the AgNPs synthesized by the five *Amaranthus* spp. was evaluated and analysed in comparison with the controls (ampicillin disc, $AgNO_3$ and the tested *Amaranthus* spp. aqueous extracts) against four pathogenic bacterial strains: *S. aureus*, *P. aeruginosa*, *E. coli*, and *K. pneumoniae*.

No antibacterial activity was detected for all the *Amaranthus* spp. aqueous extracts and low activity was noted for $AgNO_3$. However, *Amaranthus* spp. based AgNPs showed significantly larger inhibition zones for both tested Gram-negative and Gram-positive bacteria in relation to $AgNO_3$ ($P < 0.0001$). The highest antibacterial activity was observed for L-AgNPs against *E. coli* with 37 mm inhibition zone, whereas the lowest antibacterial activity was observed for B-AgNPs against *S. aureus* with 16 mm inhibition zone. *E. coli* showed the highest sensitivity to B-AgNPs, L-AgNPs, V-AgNPs, but its sensitivity to G-AgNPs and D-AgNPs was also high. No specific trend of observation was noted regarding the response of Gram-negative and Gram-positive to the biofabricated AgNPs. Generally, the antibacterial activity of the tested agents against all bacterial strains showed significant variations, as shown in Fig. 8 and Table S1. The source of variations, agents, bacterial strain, and their interaction were highly significant $P < 0.0001$ (Table S2). Detailed descriptions of the data analysis by two-way ANOVA are presented as supporting materials as Table S3 (Tukey's test) and S4 (Fisher's LSD).

## DISCUSSION

### Taxonomic impression of the morphometric features

Studies on the genus *Amaranthus* face a plenty of challenges with regard to identification and taxonomic divisions (*Assad et al., 2017*; *Iamonico, 2020a*). Starting from the study on

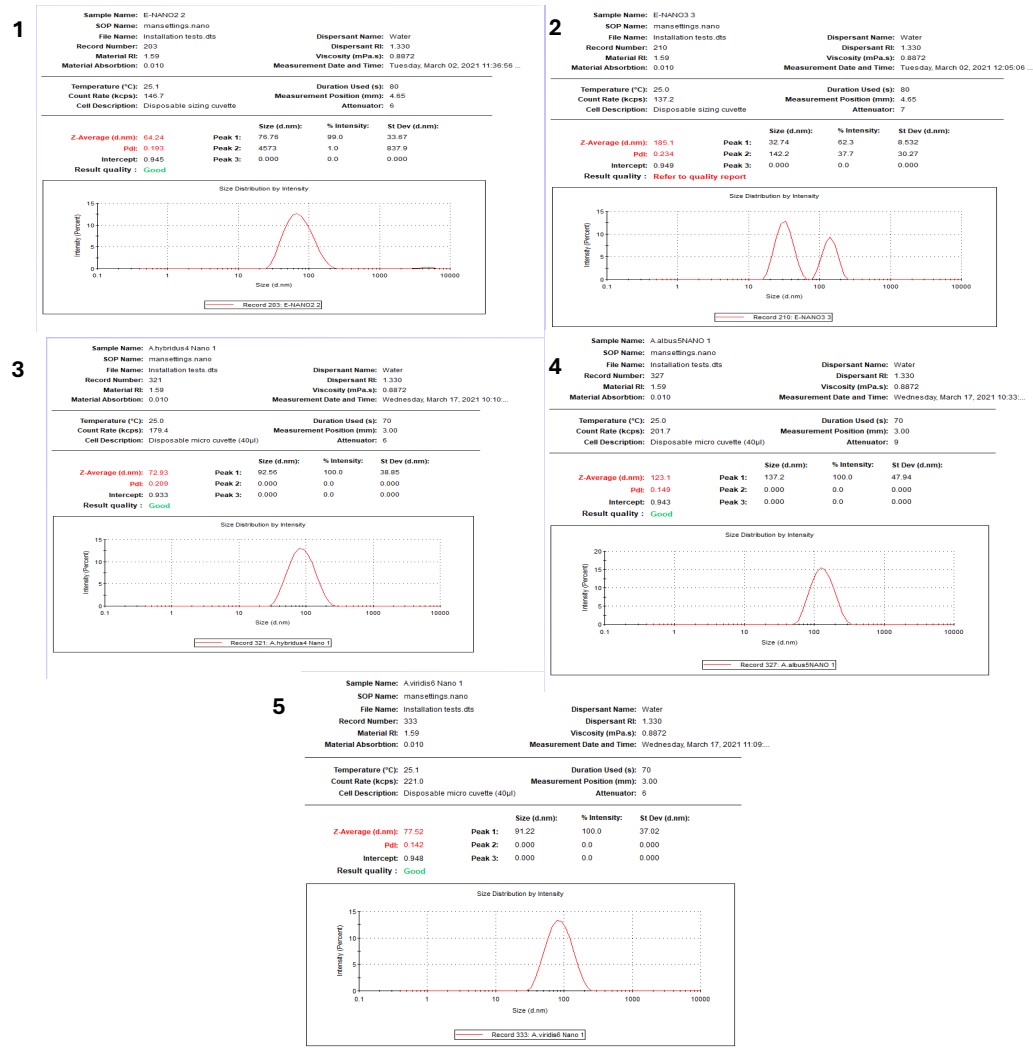

**Figure 5** **Size distribution of AgNPs synthesized from five *Amaranthus* species.** B-AgNPs (1), G-AgNPs (2), D-AgNPs (3), L-AgNPs (4), and V-AgNPs (5).

the Linnaean name (*Iamonico, 2014a*; *Iamonico, 2014b*), it was revealed that most names are considered as heterotypic synonyms (*Iamonico, 2016a*; *Iamonico, 2016b*; *Iamonico, 2020c*; *Iamonico & Palmer, 2020*). Results of the present study indicated an interesting morphological variability, especially in *A. blitoides* and high phenotypic variability (vegetative and generative characters) in *A. graecizans* and *A. blitum*. These results are in accordance with those of *Das & Iamonico, 2014*; *Iamonico (2014b)*. The main characteristic features that differentiate between the tested species were the number of the tepals, structure of the synflorescence, dehiscence/indehiscence of the fruit, and ornamentation of the fruit surface. These characteristics are valid for the description of some *Amaranthus* species (*Cappers & Bekker, 2022*; *Iamonico, 2015*).

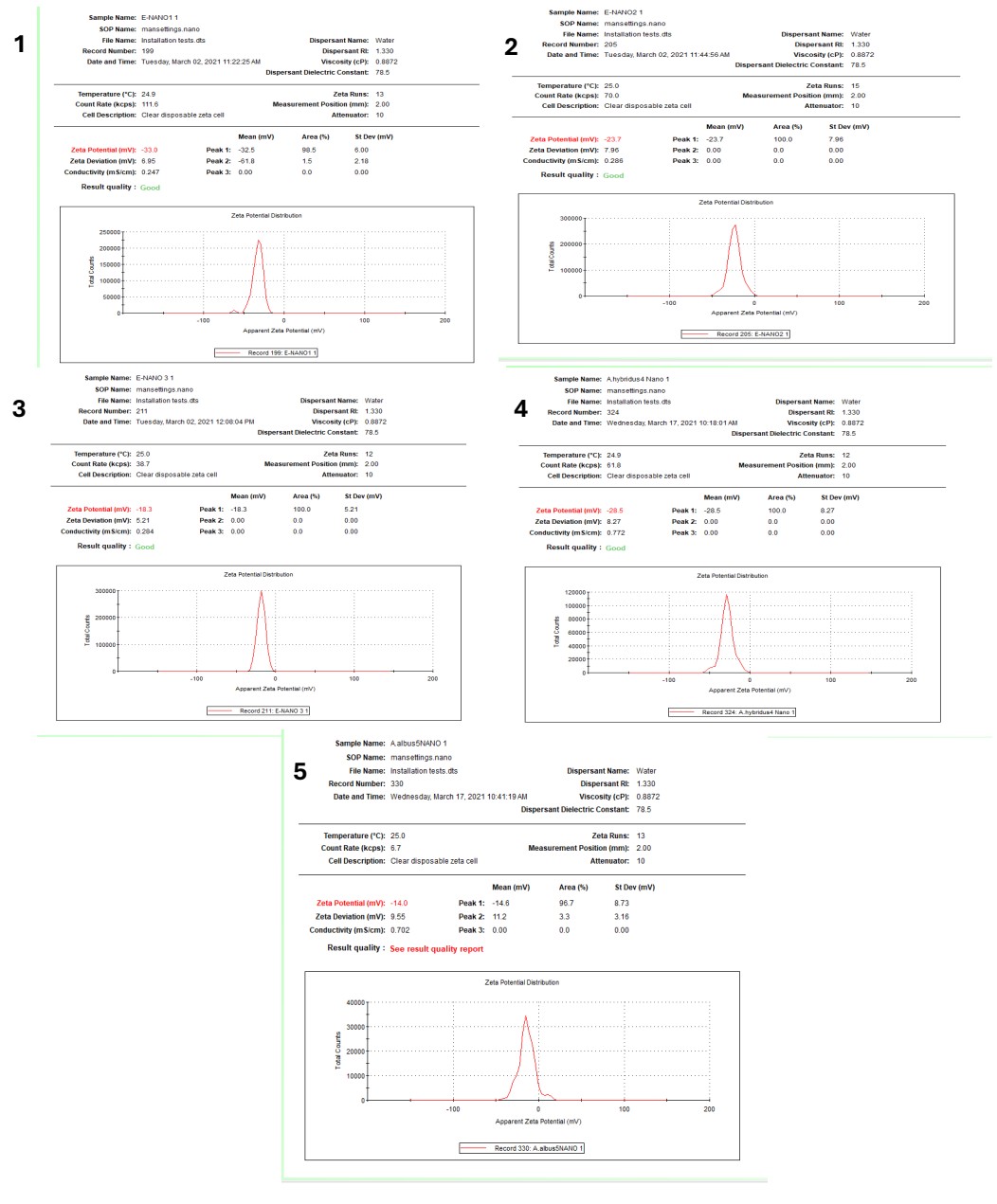

**Figure 6** **Zeta potential of AgNPs synthesized from five *Amaranthus* species.** B-AgNPs (1), G-AgNPs (2), D-AgNPs (3), L-AgNPs (4), and V-AgNPs (5).

## Nano-materials fabrication in relation to *Amaranthus* species

Conversion of metal ions to metal NPs using biological means could be mainly attributed to phytomolecules, such as alcohols or proteins and hydrocarbons that act as agents for metal ions reduction and capping the resultant metal NPs, respectively. Successful fabrication of AgNPs using *Amaranthus* species was first observed by the brown colour obtained after mixing plant extracts with AgNO₃ as typical attributes of the plasmon

**Table 1  Size, potential and polydispersity index (PDI) for the AgNPs prepared by the five *Amaranthus* species.**

|   | NPs type[a] | Size | Potential | PDI |
|---|---|---|---|---|
| 1 | B-AgNPs | 64.24 | −33 | 0.19 |
| 2 | G-AgNPs | 185.1 | −23.7 | 0.23 |
| 3 | D-AgNPs | 72.93 | −18.3 | 0.20 |
| 4 | L-AgNPs | 123.1 | −28.5 | 0.14 |
| 5 | V-AgNps | 77.52 | −14 | 0.14 |

Notes.

[a]*Amaranthus* spp. based AgNPs. B-AgNPs, G-AgNPs, D-AgNPs, L-AgNPs and V-AgNPs were the AgNPs prepared from *A. blitum* subsp. *blitum* var. *blitum*, *A. graecizans* L. subsp. *graecizans*, *A. dubius*, *A. blitoides* var. *blitoides*, and *A. viridis*, respectively.

vibration excitation in AgNPs surface (*Tripathy et al., 2010*). NPs chracterestics could be explained by *Amaranthus* species biomolecules (type and concentration) that helped in the reduction process of NPs (*Rajeshkumar & Bharath, 2017*). All prepared AgNPs were noted at 400–450 nm surface plasmon resonance (SPR) bands by UV, which is the known range for well dispersed AgNPs in solution with unique features (*Korbekandi & Iravani, 2012*; *Yamada et al., 2018*). Uniformity of AgNPs in size and shape could be predicted from the shape of SPR, where spherical NPs is expected due to single band (*Shinde et al., 2013*). *A. dubius* was previously used for AgNPs fabrication and showed SPR band occuring at 420 nm (*Firdhouse & Lalitha, 2012*; *Sigamoney et al., 2016*), which is comparable to the current findings for the five *Amaranthus* species.

*Fourier transform infrared spectroscopy*

In comparison with the five extracts and NPs tested, it was clear that all spectra showed similar main troughs of 3,250–3,320 and 1,630–1,640 cm$^{-1}$, which are mostly assigned to functional groups like $^-$NH or $^-$OH and carbonyl stretching in proteins, respectively. Therefore, it could be assumed that the NPs were capped with polyphenolic compounds and protein from the plant extracts, which could be responsible for the reduction process of Ag ions to AgNPs. FTIR spectra values shifted slightly between the plant extract and NPs for each species indicating that the same functional groups were consumed for NPs fabrication as reducing and capping agents. Further, the FTIR peaks displayed similar functional groups for all tested AgNPs suggesting that the phytochemicals that capped the surface of the biosynthesized AgNPs were similar, but they differed only in their concentrations. Slight variations between the species in the FTIR data of NPs could be related to phytomolecules types and concentrations in each species that might affect the NPs size, potential and activity. Therefore, variations in AgNPs properties could be mainly affected by the plant metabolites capping the NPs that were noted in the FTIR spectrum. Further, EDX indicated the presence of C and O beside Ag ions, thus confirming the FTIR data (*Ali et al., 2015*).

*Hydrodynamic size and surface area analysis*

Significantly smaller mean size particle diamater of B-AgNPs, D-AgNPs and V-AgNPs compared with that of G-AgNPs and L-AgNPs could possibly be related to the biomolecules from each extract that might be species- specific, since various phyto molecules may

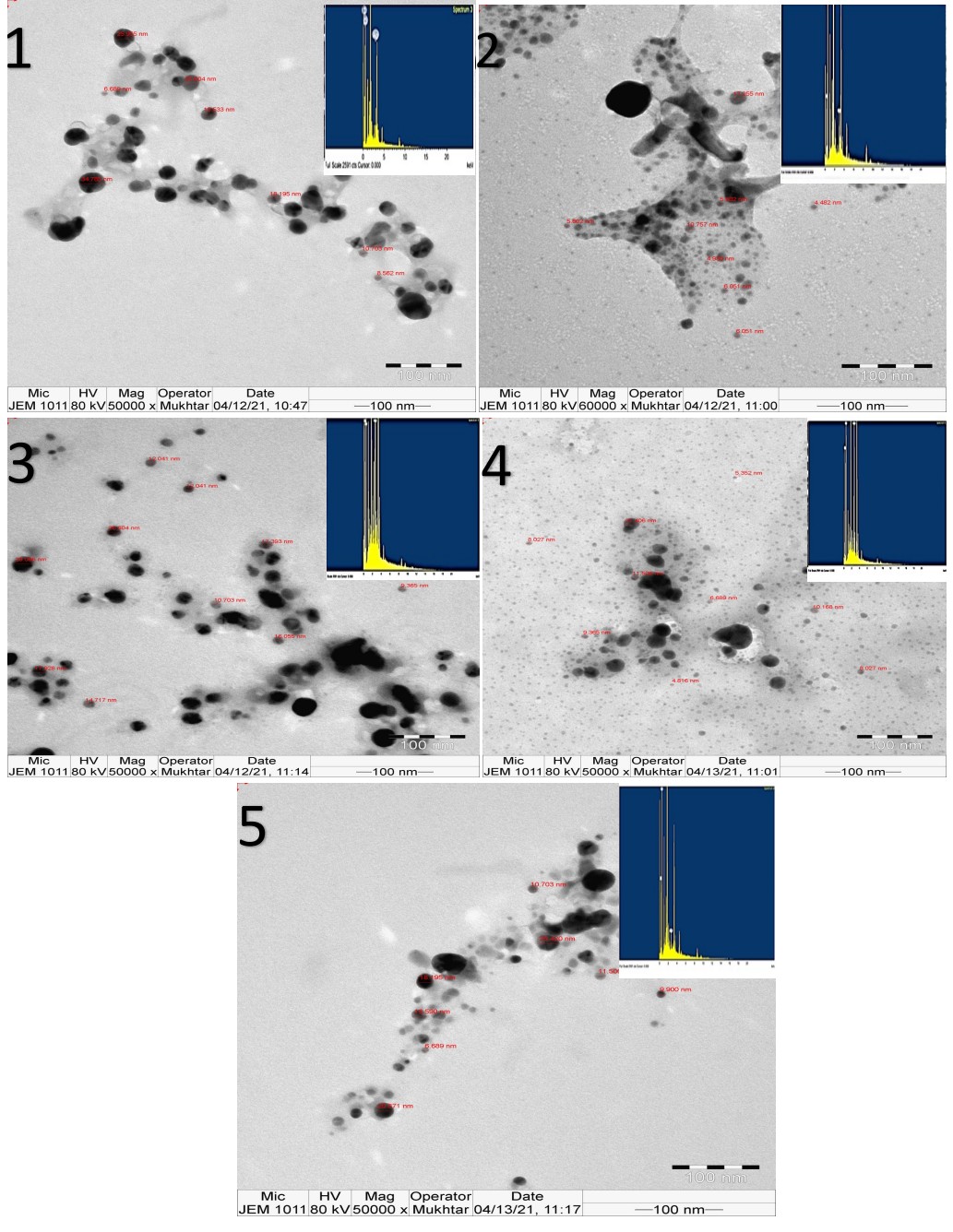

**Figure 7** TEM image and EDX presenting morphological features of NPs synthesized from five *Amaranthus* species showing surface morphology and quantitative analysis of silver atoms, carbon and oxygen for B-AgNPs (1), G-AgNps (2), D-AgNPs (3), L-AgNPs (4), and V–AgNPs (5).

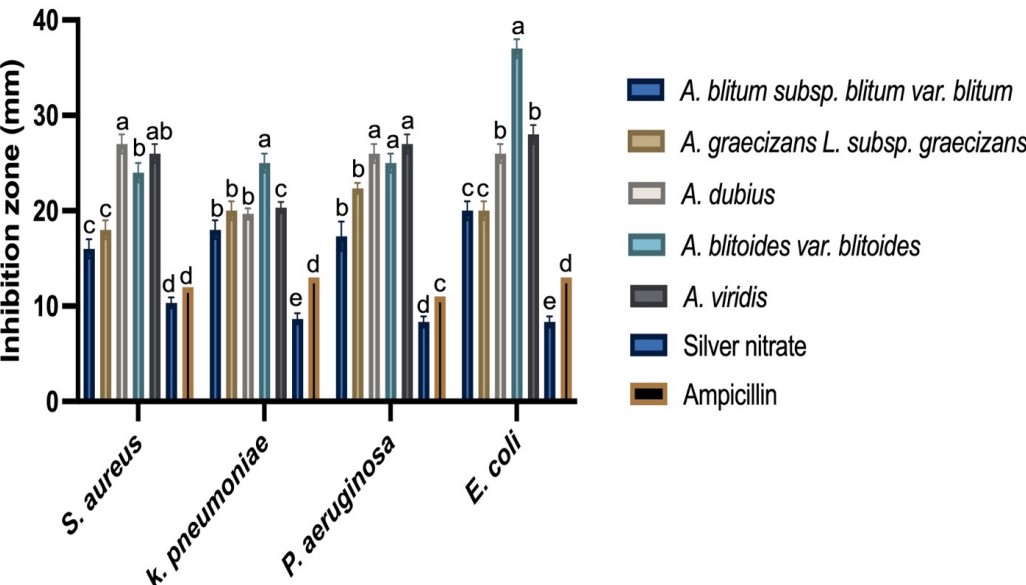

**Figure 8** **Anti-bacterial activity of AgNPs prepared by the five *Amaranthus* species as inhibition zones (mm) on nutrient agar inoculated with different bacterial strains.** Ampicillin, and AgNO₃ were included as controls. The data presented are mean values from three replicates and different letters indicated the significant variations among the AgNPs tested against each bacterial species.

influence the AgNPs synthesis, size, shape and yield, as explained by *Sigamoney et al. (2016)*, who investigated different plant parts of *A. dubius*. The relationship between phytoconstituents and AgNPs size was well documented by *Nguyen et al. (2020)*, who suggested that higher concentrations of alkaloids and saponins in *Momordica charantia* when compared with *Psidium guajava* could be the reason for *M. charantia* in providing smaller NPs.

The size of B-AgNPs, D-AgNPs and V-AgNPs were lower than 80 nm compared to G-AgNPs and L-AgNPs that indicated more than 120 nm size which might be related to the nature of the capping agent since it has great role in AgNPs production. Accordingly, it might be suggested that higher concentration of the phtochemicals leading to higher reducing and capping ability that would have suppressed AgNPs agglomeration leading to the small size NPs. Therefore, *A. blitum* subsp. *blitum* var. *blitum*, *A. dubius* and *A. viridis* could be categorized as one group and *A. graecizans*. subsp. *graecizans* and *A. blitoides* var. *Blitoides* is an another group since each group has its own potential to provide specific NPs size.

Furthermore, all plant species provided NPs with negative zeta potential, which could be a good indication for contribution of phyto-molecules that have negatively charged functional groups around AgNPs. Such phenomenon indicates good stability of AgNPs due to electrostatic repulsion between the NPs (*Sujitha & Kannan, 2013*). This study synthesized NPs that varied in size and potential. These characteristics are affected by many factors including NPs prepartion conditions; but, the conditions in synthesizing the NPs

in this study were the same, therefore, these variations could be mainly attributed to the plant constituents types and concentrations due to their role in reduction and stabilization processes.

### Morphology and size distribution analysis using transmission electron microscopy with energy-dispersive X-ray spectroscopy

In the current investigation, TEM analysis indicated the spherical shape for the synthesized AgNPs, regardless of the plant species, which was previously observed for AgNPs prepared by *A. dubius* (*Firdhouse & Lalitha, 2012*). Regarding Ag weight (%) detected by EDX-and the AgNPs size, a negative correlation is expected since larger particles size is supposed to have more Ag ions. However, it was not the case in the current findings; this might be due to the significant variations among the species tested, as higher AgNPs yield is phytochemically dependent (*Hemath Naveen et al., 2010*).

## Antibacterial activity of the biosynthesized AgNPs

Higher plants are capable of producing a large number of natural products, which are known as secondary metabolites (*Bonanomi, Vinale & Scala, 2009*). For example, *Thymus vulgaris, Origanum vulgare, Rosmarinus officinalis*, and *Cinnamon* comprise compounds, such as thymol, carvacrol, camphor, and cinnamaldehyde, that exhibited antimicrobial activity against many pathogenic bacterial species, such as *Salmonella enterica* Typhimurium, *Staphylococcus aureus*, and *Listeria monocytogenes* (*Guo et al., 2020*; *Pesavento et al., 2015*; *Rao, Chen & McClements, 2019*). Also, plants exhibit diverse organic compounds like phenols and oxygenated terpenoids, which could enhance their antimicrobial activity (*Rao, Chen & McClements, 2019*). Plant terpenoids may increase membrane permeability by disturbing the fatty acid composition of the bacterial cell membrane, leading to leakage of cellular contents (*Trombetta et al., 2005*). Phenols may also lead to structural and functional damages of bacterial cell membranes (*Rao, Chen & McClements, 2019*). However, the currently tested *Amaranthus spp. had no direct antibacterial effect against the tested strains. This suggests that the concentrations of the plant extracts used was not enough for bacterial suppression.* But, the biofabricated NPs by these extracts inhibited the growth of all tested bacterial strains, thus indicating the higher efficiency of the fabricated AgNPs in bacterial suppression in relation to their individual constituents (AgNO$_3$ and plant extracts), which might be explained by the fact that small size particles are better in penetrating the bacterial cells.

To the best of our knowledge, the current study is the first comparative investigation that sought out the characteristics and biological activities of AgNPs fabricated using five *Amaranthus* spp. in Saudi Arabia. From the current findings, *E. coli* was the most sensitive tested pathogen by most of the AgNPs prepared by *Amaranthus* spp., especially L–AgNPs. In a previous study, AgNPs and alcoholic extract of *Amaranthus retroflexus* had higher effect against the Gram-negative bacteria (*P. aeruginosa*) than the Gram-positive bacteria, *S. aureus* (*Rasi-Bonab et al., 2018*). Moreover, the green synthesis of AgNPs using *A. gangeticus* Linn leaf extract revealed NPs with antimicrobial properties against *Bacillus subtilis*, and *Shigella flexineri* (*Kolya et al., 2015*). The antibacterial properties of AgNPs could rely on their ability to enter the microbial cells, damage the DNA, and

lose cell contents (*Manivasagan et al., 2013*). Furthermore, it could also be related to the development of reactive oxygen species that interact with glycoproteins on the bacterial cell wall, which facilitate their entry into the cytoplasm (*Osonga et al., 2020*). Additionally, the small NPs size increases their contact surface with the outer membrane of the bacterial cells, thus enhancing their antibacterial effect (*Nakagawa et al., 1999*). NPs characteristics, such as antimicrobial activity, are also shape- dependent since triangular NPs are better than spherical and both are better than rod shape NPs (*Pal, Tak & Song, 2007*).

Generally, the investigated *Amaranthus* species revealed AgNPs with various characteristics that were noted *via* UV-absorption, DLS data, TEM and EDX analysis, FTIR as well as antibacterial activity. No special trend of observations was noted regarding the antibacterial effect of the prepared NPs, which might be explained by the fact that NPs chracterestics slightly varied, especially in size and potential, however, the NPs effect could also be species- specific.

## CONCLUSION

The current study indicated that different *Amaranthus* species fabricated NPs that have varied characteristics specifically size, in addition to their morphological evidence. Bio fabricated NPs indicated variations in properties in terms of size, potential, production level, as well as antibacterial effect. Size and potential variability of the NPs might be explained by types of plant constituents and concentrations that helped in the reduction, capping and stabilization processes of NPs. Slight variations were noted in the FTIR spectra, which suggest that the phyto-molecules detected in each species could be slightly different in terms of type and concentration, which in turn could be the main effective factors in phytofabrication of NPs.

## ACKNOWLEDGEMENTS

Dr. Yahya S. Masrahi and Dr. M. Remesh, Herbarium, Department of Biology, Faculty of Science, Jazan University, helped in samples collection.

### Funding

This research has been supported by Princess Nourah bint Abdulrahman University Researchers Supporting Project number (PNURSP2024R187), Princess Nourah bint Abdulrahman University, Riyadh, Saudi Arabia. The funders had no role in study design, data collection and analysis, decision to publish, or preparation of the manuscript.

### Grant Disclosures

The following grant information was disclosed by the authors:
Princess Nourah bint Abdulrahman University Researchers Supporting Project: PNURSP2024R187.

## Competing Interests

Duilio Iamonico is an employee of Garden of the Apennine Flora of Capracotta, Italy.

## Author Contributions

- Walaa A. Hassan conceived and designed the experiments, performed the experiments, prepared figures and/or tables, and approved the final draft.
- Afrah E. Mohammed performed the experiments, analyzed the data, prepared figures and/or tables, authored or reviewed drafts of the article, and approved the final draft.
- Najla A. AlShaye performed the experiments, authored or reviewed drafts of the article, and approved the final draft.
- Hana Sonbol performed the experiments, authored or reviewed drafts of the article, and approved the final draft.
- Salma A. Alghamdi performed the experiments, authored or reviewed drafts of the article, and approved the final draft.
- Duilio Iamonico performed the experiments, authored or reviewed drafts of the article, and approved the final draft.
- Shereen M. Korany conceived and designed the experiments, performed the experiments, prepared figures and/or tables, and approved the final draft.

## Data Availability

The raw data are available in the Supplemental Files.

## Supplemental Information

Supplemental information for this article can be found online at http://dx.doi.org/10.7717/peerj.16708#supplemental-information.

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
