# Peer review of "Characterization of Amaranthus species: ability in nanoparticles fabrication and the antimicrobial activity against human pathogenic bacteria"

_PeerJ, doi:10.7717/peerj.16708_

## Round 0.1 · original submission · Major Revisions

Please respond to all points. In particular, please improve the methodology and discussion section.

Reviewer 1 ·

Basic reporting

1. It’s noted that your manuscript needs careful editing paying particular attention to English spelling and sentence structure so that the study are clear to the readers.

Experimental design

1. The authors should conduct qualitative and quantitative analysis of the five Amaranthus species extracts.

Validity of the findings

1. In addition, the authors said nanoparticles approach was used to distinguish the five Amaranthus species, but I can’t read any useful information to support this opinion. Please clarify how we can distinguish these five Amaranthus species by nanoparticles.

Additional comments

1. In 2.2.3.3, the title is not correct. It should be changed into “Hydrodynamic Size and Surface Potential Analysis”. In 3.2.3, the title also needs to be changed.
2. All the figures need to be modified, because they are not clarity enough.
3. In Figure 8, the color of the legend for Ampicillin need be changed into another color which can distinguish from A. graecizans L. subsp. graecizans.

Reviewer 2 ·

Basic reporting

1- Title is ambiguous with inappropriate words like “discrimination”
2- There is lack of clear, unambiguous, professional English language throughout the MS. English editing strongly recommended.
3- In abstract, abbreviations used at first mentioned should be described fully.
4- Sentences are not clear as required for an international publication.
5- Introduction is direction less. Need major revisions to adhere to the topic under investigation.
6- No rationale provided in intro.
7- in vitro always italicized.
8- line # 140, CFU expression needs to be checked again.
9- Scientific names need to be with full genus name at 1st mention like line 232-233.
10- Table 2, authors should explain why there was no SE in positive control?

Experimental design

1- The authors have adopted acceptable experimental design. But there is need for proper statistical analyses.

Validity of the findings

1- Fig, 2. Pics not captured to scale.
2- Fig. 8 looks duplication of data as in Table 2 also?
3- Research question not well defined, relevant & meaningful.
4- The authors concluded as “Current study recommended using the nanobiotechnology approach as taxonomic plant tool since varied plant species of the same genus displayed varied NPs characteristics and biological activities in addition to their morphological evidence”.
In my opinion, antibacterial activity can not be used as a tool for taxonomic characterization because the activity may change with the passage of time and with varying bacteria as some bacteria may develop resistance against some antibacterial chemicals. Moreover, different Amaranthus species growing in different locations will have a varied biochemical profile that is important to exhibit consistent results. The time/stage of maturity of any plant is also very important because bioactive chemical compounds present in any plant are dependent on the growth stage of any plant. Based on these facts, I suggest to decline this MS.

·

Basic reporting

Line 95 spelling “easch”
Line 96 and other places ml should be mL
Line 101 AgNO3 subscript in formula
The authors have used Amaranthus spp. so need to justify the rationale behind using this species and its significance. The novelty of this method to be discussed.



Table 1 gives variation in size of AgNPs, the authors should elaborate upon the reason behind this in the paper, only explained briefly in the conclusion part.

Experimental design

The biological activity against these bacterial strains is already reported, the authors may investigate activity against more strains and some resistant strains

Validity of the findings

no comment

---

## Round 0.2 · Minor Revisions

The authors have improved their manuscript. However, some further improvement is required. Cruilayy, please note the comment of a Reviewer in your revision taht "the MS can be published if the authors just report the morphological identification and bifocality of nanoparticles excluding the claim of nanoparticles as a tool for identification of species".

Reviewer 1 ·

Basic reporting

no comment

Experimental design

no comment

Validity of the findings

no comment

Reviewer 2 ·

Basic reporting

1- The authors have greatly improved the MS and its English.

2- There is no rationale provided in introduction to justify the problem under discussion.

Experimental design

1- No statistical analysis in fig 8, although the I have pointed out in my previous comments.

Validity of the findings

1- I am repeating my previous comments as the authors did not provide logical reply. “The authors concluded as “Current study recommended using the nanobiotechnology approach as taxonomic plant tool since varied plant species of the same genus displayed varied NPs characteristics and biological activities in addition to their morphological evidence”.
In my opinion, antibacterial activity can not be used as a tool for taxonomic characterization because the activity may change with the passage of time and with varying bacteria as some bacteria may develop resistance against some antibacterial chemicals. Moreover, different Amaranthus species growing in different locations will have a varied biochemical profile that is important to exhibit consistent results. The time/stage of maturity of any plant is also very important because bioactive chemical compounds present in any plant are dependent on the growth stage of any plant. Based on these facts, I suggest to decline this MS”.
Moreover,
The authors failed to provide convincing evidence/argument in favor of using nano technique to be used in taxonomy of Amaranth. The authors declared in section 3. Results 3.1. Morphology that the amaranth species varied greatly on the basis of morphological characteristics so, using nano technique will not be a useful addition to resolve taxonomy issues. Metabolite profiling can be used but with its own limitations. However, nano technique may be considered where morphological identification is confusing. So, my previous comments regarding use of nano technique in taxonomy remains intact. The authors used latest expensive techniques to describe the nanoparticles however, DNA molecular identification is still the most authentic technique with less cost.
The MS can be published if the authors just report the morphological identification and bifocality of nanoparticles excluding the claim of nanoparticles as a tool for identification of species.

Additional comments

The MS can be published if the authors just report the morphological identification and bifocality of nanoparticles excluding the claim of nanoparticles as a tool for identification of species.

---

## Round 0.3 · Minor Revisions

I regret to submit that Authors couldn't justify their claim about application of nanoparticles as a tool to differentiate plant species. Please provide concrete logic to justify your claim. Otherwise, remove such statements.

Reviewer 2 ·

Basic reporting

The authors are stil claining nanoparticles as a tool to differentiate plant species as indicated in the MS, although they agreed in response letter that they are not claiming.
From abstract;
Varied characteristics for the obtained AgNPs may indicate the ability of the nano techniques in differentiating among Amaranthus species in relation to their phytochemical composition and concentration.
From Introduction;
The current work represents a pioneer attempt for using AgNPs to categorize the genus Amaranthus with reference to their bio-reducing and capping abilities which may offers AgNPs with varied characteristics. Such variation could be helpful in differentiating among the tested species
From conclusion;
The current study indicated that the nano biotechnology approach could be a tool for differentiating Amaranthus species since displayed NPs that varied in characteristics specifically size, in addition to their morphological evidence.

Experimental design

The authors need to mention how they performed statistical test in figure 8? Separate on each species or combined, what test was used?

Validity of the findings

NA

Additional comments

NA

·

Basic reporting

The authors has worked on the revised version.

Experimental design

Should give a tabular representation of the anti-bacterial activity and the plant extract

Validity of the findings

The observations are convincing

---

## Round 0.4 · Minor Revisions

Please improve your article as advised by the Reviewer.

Reviewer 2 ·

Basic reporting

The authors have greatly modified the MS.

Experimental design

Need statistical analyses e.g., some multiple range tests in tables and figures. ANOVA not sufficient,

Validity of the findings

NA

---

## Round 0.5 · Minor Revisions

Dear Authors,
Please verify:
L411-413= Such variations likely demonstrate the ability of nano- techniques to differentiate among the tested Amaranthus species leading to categorize them into two groups since each group has its own potential to provide specific NPs size.
or
DELETE this information

Reviewer 2 ·

Basic reporting

L 168= spell mistake tow-way
L 191= A. graecizans L. when you abbreviate genus name then there is no need for author citation. Check in all MS

L 184= In figure descriptions, full genus name is recommended. Check in all MS
L 267= Tukey's multiple comparisons test should be written as Tukey's test

L411-413= Such variations likely demonstrate the ability of nano- techniques to differentiate among the tested Amaranthus species leading to categorize them into two groups since each group has its own potential to provide specific NPs size.
DELETE this information

Experimental design

NA

Validity of the findings

L411-413= Such variations likely demonstrate the ability of nano- techniques to differentiate among the tested Amaranthus species leading to categorize them into two groups since each group has its own potential to provide specific NPs size.
DELETE this information

Additional comments

The authors need to follow the comments if they are unable to provide evidence to proof their claim.
L411-413= Such variations likely demonstrate the ability of nano- techniques to differentiate among the tested Amaranthus species leading to categorize them into two groups since each group has its own potential to provide specific NPs size.

---

## Round 0.6 · accepted · Accept

Authors have improved their manuscript and now it may be recommended for publication.

Reviewer 2 ·

Basic reporting

The authors have greatly improved the MS, however there is need to prrofread and add recent literarure showing antimicrobial activity of used plant like
Antibacterial and antioxidant activities of slender amaranth weed. Planta Daninha. 38: e020192974. https://www.scielo.br/j/pd/a/73WNXnc6QH6b3m8ynWzmwYK/?lang=en#:~:text=viridis%20leaf%20exhibited%20the%20best,carotovora.

Experimental design

OK

Validity of the findings

OK

Additional comments

NA